# Living with Floods Using State-of-the-Art and Geospatial Techniques: Flood Mitigation Alternatives, Management Measures, and Policy Recommendations

Rabin Chakrabortty, Subodh Chandra Pal *, Dipankar Ruidas, Paramita Roy, Asish Saha and Indrajit Chowdhuri

Department of Geography, The University of Burdwan, Raiganj 713104, West Bengal, India
* Correspondence: scpal@geo.buruniv.ac.in

**Abstract:** Flood, a distinctive natural calamity, has occurred more frequently in the last few decades all over the world, which is often an unexpected and inevitable natural hazard, but the losses and damages can be managed and controlled by adopting effective measures. In recent times, flood hazard susceptibility mapping has become a prime concern in minimizing the worst impact of this global threat; but the nonlinear relationship between several flood causative factors and the dynamicity of risk levels makes it complicated and confronted with substantial challenges to reliable assessment. Therefore, we have considered SVM, RF, and ANN—three distinctive ML algorithms in the GIS platform—to delineate the flood hazard risk zones of the subtropical Kangsabati river basin, West Bengal, India; which experienced frequent flood events because of intense rainfall throughout the monsoon season. In our study, all adopted ML algorithms are more efficient in solving all the non-linear problems in flood hazard risk assessment; multi-collinearity analysis and Pearson's correlation coefficient techniques have been used to identify the collinearity issues among all fifteen adopted flood causative factors. In this research, the predicted results are evaluated through six prominent and reliable statistical ("AUC-ROC, specificity, sensitivity, PPV, NPV, F-score") and one graphical (Taylor diagram) technique and shows that ANN is the most reliable modeling approach followed by RF and SVM models. The values of AUC in the ANN model for the training and validation datasets are 0.901 and 0.891, respectively. The derived result states that about 7.54% and 10.41% of areas accordingly lie under the high and extremely high flood danger risk zones. Thus, this study can help the decision-makers in constructing the proper strategy at the regional and national levels to mitigate the flood hazard in a particular region. This type of information may be helpful to the various authorities to implement this outcome in various spheres of decision making. Apart from this, future researchers are also able to conduct their research byconsidering this methodology in flood susceptibility assessment.

**Keywords:** flood hazard; Kangsabati river; RS-GIS; subtropical climate; machine learning model

## 1. Introduction

In the world, flood is the most widely known and devastating hazard among various hydro-meteorological hazards [1–3]. During flood time, severe human conditions and socio-economic damage are caused. Floods occurred about 150,061 times worldwide as per a report by the "United Nations Office for Disaster Risk Reduction Statistical Data" (UNISDR). From 1995 to 2015, around 157,000 people died because of this hazard, which is responsible for 11.1% of natural disaster fatalities [4]. According to Nicholls et al., 1999 [5], worldwide, floods have an impact on almost 200 million people each year. According to "climate change forecasts", based on changes in land use patterns and an increase in "population", flood occurrence rates and severity are predicted to worsen by 2050 and may result in significant losses [6]. However, the Great Himalayan Glaciers and the South Asian

Subcontinent remain the most vulnerable. While wet weather is still the primary culprit for floods in the area, glaciers and rising temperatures pose a long-term problem [7,8]. At a rate of 40 percent, floods are the region's most frequent natural calamity. In addition, 9.6 million individuals in Bangladesh, India, and Nepal have been impacted by urban floods due to solid waste management and declines in urban vegetation. Of them, 6.8 million are from India alone. 9.6 million people have been impacted by the flooding in South Asia, making this an increasingly urgent humanitarian disaster [7]. Therefore, it is vitally necessary to conduct research on flood susceptibility at the basin size on the regional level in order to implement management measures to control flood events. Extreme precipitation events are expected to occur more often across the world in the near future, according to the IPCC's Sixth Assessment Report. In addition, it is anticipated that the natural water retention by land use would diminish as urban land uses increase in the future. As a result, an increase in the frequency and detrimental effects of flood occurrences is anticipated.

Like other natural disasters ("landslides, volcanoes, or earthquakes, flood"), disasters occur more frequently associated with a broader impact [9]. Storm surges, severe or sudden snowmelt, a rising water level from melting snow, and overtopped embankments are all potential causes of flooding [10]. Flooding may also be caused by other disasters and situations, such as landslides or tsunamis caused by an earthquake. Scientists anticipate that as a result of climate change, the world will see an increase in the frequency and intensity of floods and droughts worldwide, as well as an increase in the uncertainty around coastal flooding due to rising sea levels. A basin may flood due to the fast population growth brought on by urban development. Flood disasters are frequently more prevalent in regions with higher population densities, more agricultural areas, or denser river networks [11]. As a result, determining the carrying capacity of river basins at risk of flooding and managing watersheds on a worldwide basis is crucial. Floods are a recurring natural occurrence that cannot be avoided. However, it is possible and desired to lessen their negative effects, particularly close to important infrastructure and residential areas. The expensive floods at the start of the twenty-first century is one of the most influential aspect which is directly influences the livelihood of the people [12]. For logical flood risk management and the prioritizing of flood-prone regions, it is thus required to identify the critical infrastructure (CI) and social infrastructure (SI) that is exposed to flooding. The capacity to analyze the vulnerability of infrastructure to flooding on a national scale is now possible because to the expanding availability of EO data and the widespread usage of GIS. The regional differences of CI flood exposure that were uncovered in this study potentially reflect various flood management techniques applied in different areas and sectors. These flood management techniques are completely reliant on the decision makers' consciousness of and responsiveness to flood hazard, the operations and maintenance conditions of the CI facilities, and the local socio-environmental conditions [13].

There have been many approaches used to evaluate and identify flood-prone locations, but lately, the machine learning model incorporating geospatial techniques has gained popularity. According to the literature, earlier studies have included "MCDA approaches, including the AHP and the expert scoring system". The accuracy of these approaches is based on professional understanding. Numerous "physically based models" (like "VIC and MIKE") and as well as different hydrological models are used to study floods at the continental and global levels [14]. Floods are studied using a variety of "physically oriented models" (such as "VIC and MIKE") at global level. The machine learning models build a connection between the frequency of floods and the explanatory elements from the historical flooding data, avoiding the subjective weight assessment. The benefit of ML models became clearer in light of the complexity of the world's floods, which may include a big number of model parameters, repeated model debugging, and long computation durations. On the basis of these advantages, it can be stated that the LSTM network has the ability to enhance residual error characterization, allowing reliable probabilistic predictions as well as predictive inferences for variables that are difficult to detect [15]. In order to support decision-makers and hydrologists, hydrological models were created to evaluate

the effects of climate change as well as to analyze, comprehend, and look at potential approaches to sustainable water management [16].

A specific watershed was usually the subject of the "flood risk assessment", which was based on "machine learning methods". For instance, Tehrany et al. used "SVM" with a variety of "kernel types" in "flood susceptibility zone mapping (FSZM)" in identifying flood-prospected areas. The input samples in the training section are utilized as a training dataset for flood occurrence or non-occurrence points in this machine-learning model. In actuality, the characteristics of the whole basin, rather than a single sampling site, are what cause floods to occur. Additionally, it is essential to assess the "flood risk" for worldwide watersheds since these evaluations have rarely used machine learning approaches. The application of a machine learning algorithm in a GIS environment has not been done by the researchers considering the maximum possible conditioning parameters. Here we are trying to fill this gap by considering the maximum possible parameters in the overall Kangsabati river basin for flood susceptibility assessment.

## 2. Materials and Methods

### 2.1. Study Area

Kanghasabati river is entirely positioned within four districts ("Bankura, Purulia, East Midnapore, and West Midnapore") by covering 4265 km$^2$ and lies between 21°45′ N to 23°30′ N latitudes and longitudes from 85°45′ E to 88°15′ E (Figure 1). The highest precipitation occurs in the wet monsoon season and reaches 1650 mm (June to September). Summersare very hot in the months of May–June, and temperatures to 45 °C characterize the sub-tropical type of climate. The natural vegetation along with agriculture is affected by the dry and very hot summer with evaporation. Granite gneiss, Archean rock formations, laterite, and alluvial deposits in the lower part together develop the geomorphological and geological conditions of this selected study region. Soil, drainage, geology, and geomorphic landscape both establish a strong metamorphic correlation here.

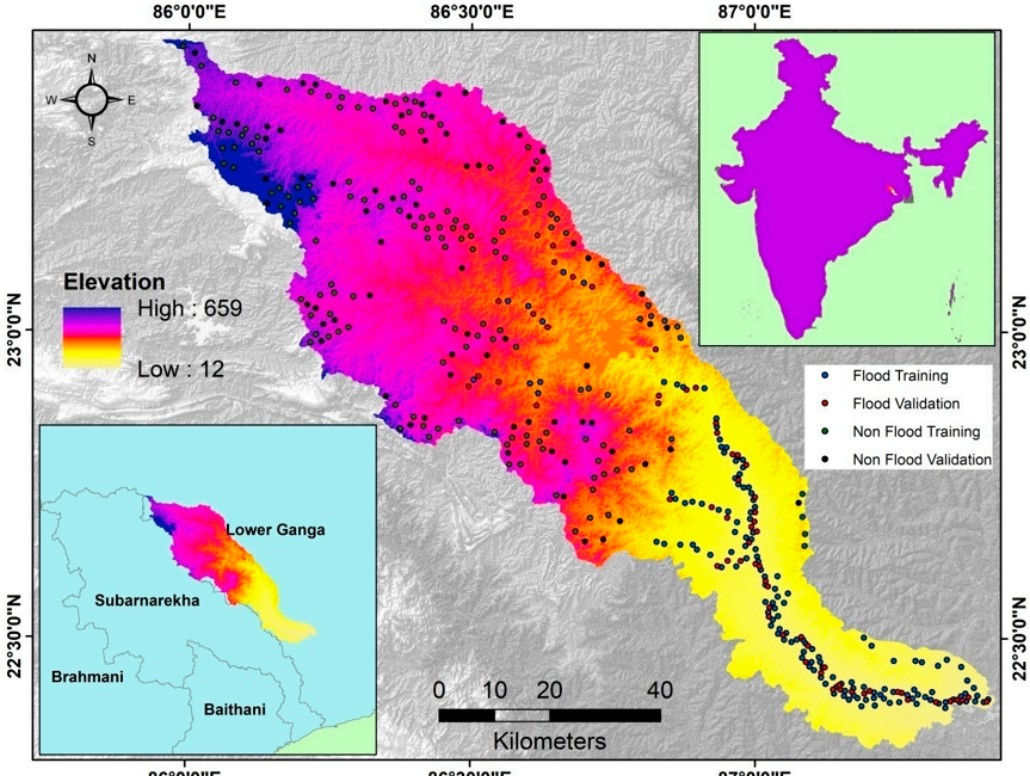

**Figure 1.** Location map of the Kangsabati river basin.

### 2.2. Methodology

The machine learning model (SVM, ANN, and RF) through the GIS platform for flood susceptibility mapping followed various steps (Figure 2): 1. The GIS database is used for a selection of conditioning parameters. 2. Preparation of 'flood inventory mapping (FIM)' with these collections. 3. Establish the flood susceptibility map using SVM, ANN, and RF models with a machine learning technique 4. With the help of a multi-collinearity test, investigation of the most suitable condition factors for flood occurrences. 5. Validated the established 'susceptibility map'. The selection of causal parameters was performed with the help of existing literature [17–20]. Then, the flood susceptibility assessment was done with the help of SVM, ANN, and RF machine learning algorithms. The spatial outcome of flood susceptibility assessment is shown in the GIS environment. The natural breaks classifier in the GIS environment was considered for classifying the flood susceptibility raster in different qualitative classes. A data categorization technique called the "Jenks Natural Breaks Classification" (or optimization) system seeks to maximize the grouping of a collection of values into "natural" classes. The best class range discovered "naturally" in a data set is known as a natural class. A class range in a data set is made up of objects having comparable qualities that constitute a "natural" group [21].

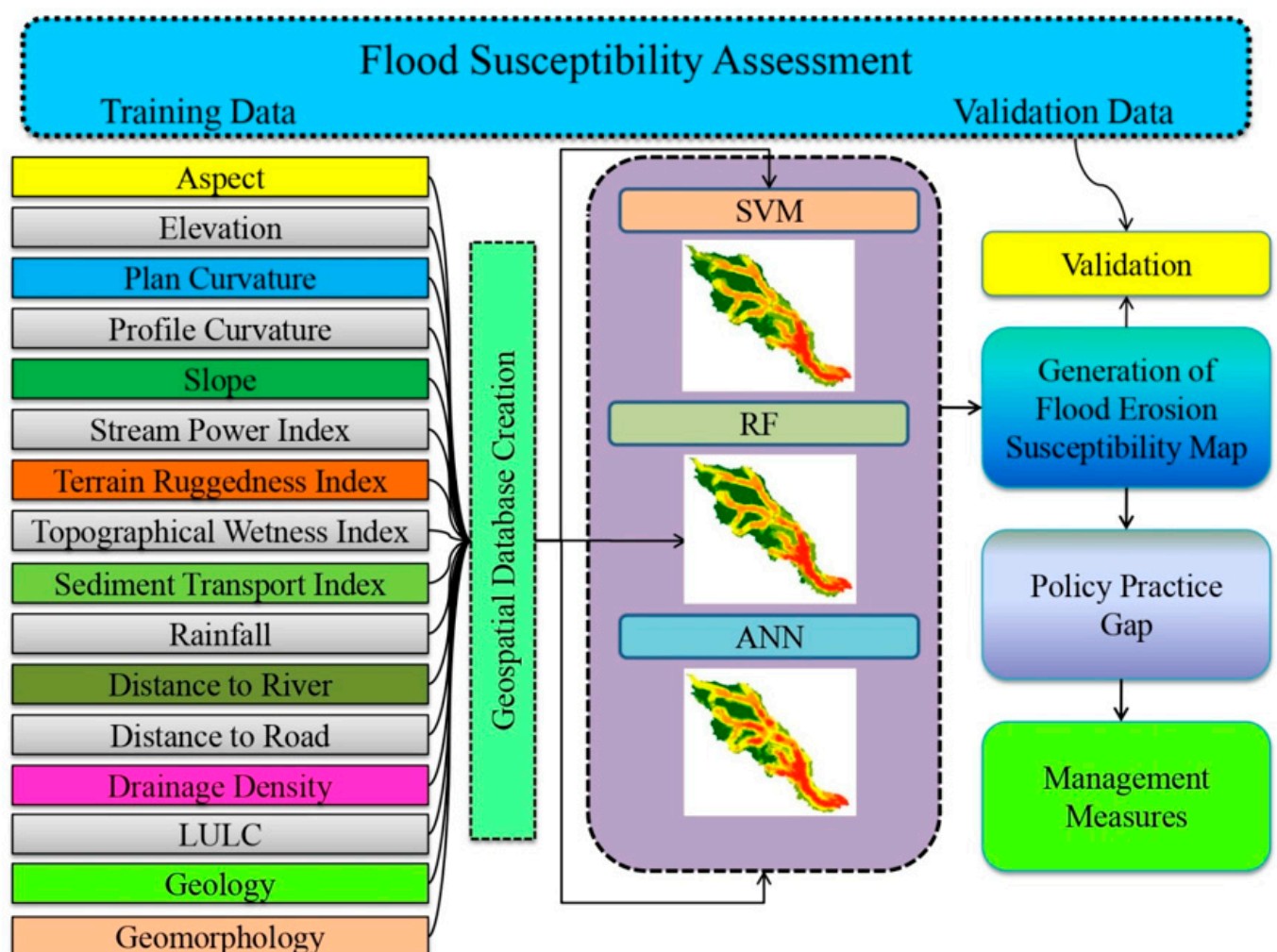

**Figure 2.** Methodology flowchart of this study.

### 2.3. Flood Inventory Mapping

The creation of inventory maps is the foremost and essential aspect for future disaster assessment [22]. An inventory map of the research region was created using the local field survey report, and previous records from different organizations were used to find out the flooded areas based on historical flood data. Numerous flooded places were arbitrarily split into two different phases, namely training (70%) and validation (30%), respectively, after the application of three models (SVM, ANN, and RF). The mentioned models selected for flood susceptibility assessment were based on the previous existing literature related to flood susceptibility research [23–26].

### 2.4. Causatives Parameter to Flood

Causative elements must be established as independent variables in order to create an FSM [27]. All of the contributing variables are not employed in all research areas since they may not have an influence in other areas, as claimed by Kia et al., (2012) [28]. In previous literature different factors such as "drainage distance, altitude, land use pattern, drainage density, water depth, rainfall and soil characteristics" has been considered [29–32]. The investigation of flood behavior is important for receiving considerable attention; on the other hand, the current focus is to establish flood susceptibility mapping by highlighting the flood causal factors along with geospatial techniques in a region. In this study, "aspect, elevation, plan curvature, Profile Curvature, slope, SPI, TRI, TWI, STI, rainfall, distance to a river, distance to road, drainage density, geology, LULC, geomorphology" were compared with previous flood related studies (Figure 3) [28,33,34]. According to Tehrany et al., 2019 [35], the topography of any region plays a significant role in "flood affected area" identification (FSI)'. It has direct effects on the speed of runoff [28], slope steepness, accelerates runoff velocity, and decreases interception rates [36]. The distance river distance from the adjacent settlement area has a significant impact on the flood's size. Flooding is influenced by drainage density, with higher drainage densities resulting in poorer interception rates and hence higher runoff concentration [37,38]. Land use has an impact on several aspects of hydrology, such as runoff, evapotranspiration of a region, and interception of water to percolation process. Less vegetation causes more flooding, whereas more vegetation causes less flooding. The features of the harmful relationship between "flood events" and "vegetation cover" are as follows.

### 2.5. Methods

#### 2.5.1. Multicollinearity Assessment

In the case of "FSM", MC assessment has emerged as a most prominent and widely used technique that is specifically used to delineate the interdependence among all adopted conditioning parameters. Previous studies have shown that if two or more variables are found interdependent, then the accuracy of the derived result significantly decreases [39]. Thus, it is an essential step toward susceptibility mapping [40]. Therefore, different scholars [35,41] have applied this tool in the identification of linearity among "flood causative factors". Generally, two indices are used for MC analysis such as "variance inflation factor (VIF)" and "Tolerance (TOL)"; the VIF value should be less than 10, and the TOL value greater than 0.1, which defines the no collinearity issues in research work. The MC test was performed by the following equations:

$$TOL = 1 - R_J^2 \tag{1}$$

$$VIF = \frac{1}{TOL} \tag{2}$$

where "$R^2$ demonstrates the coefficient of regression" [42].

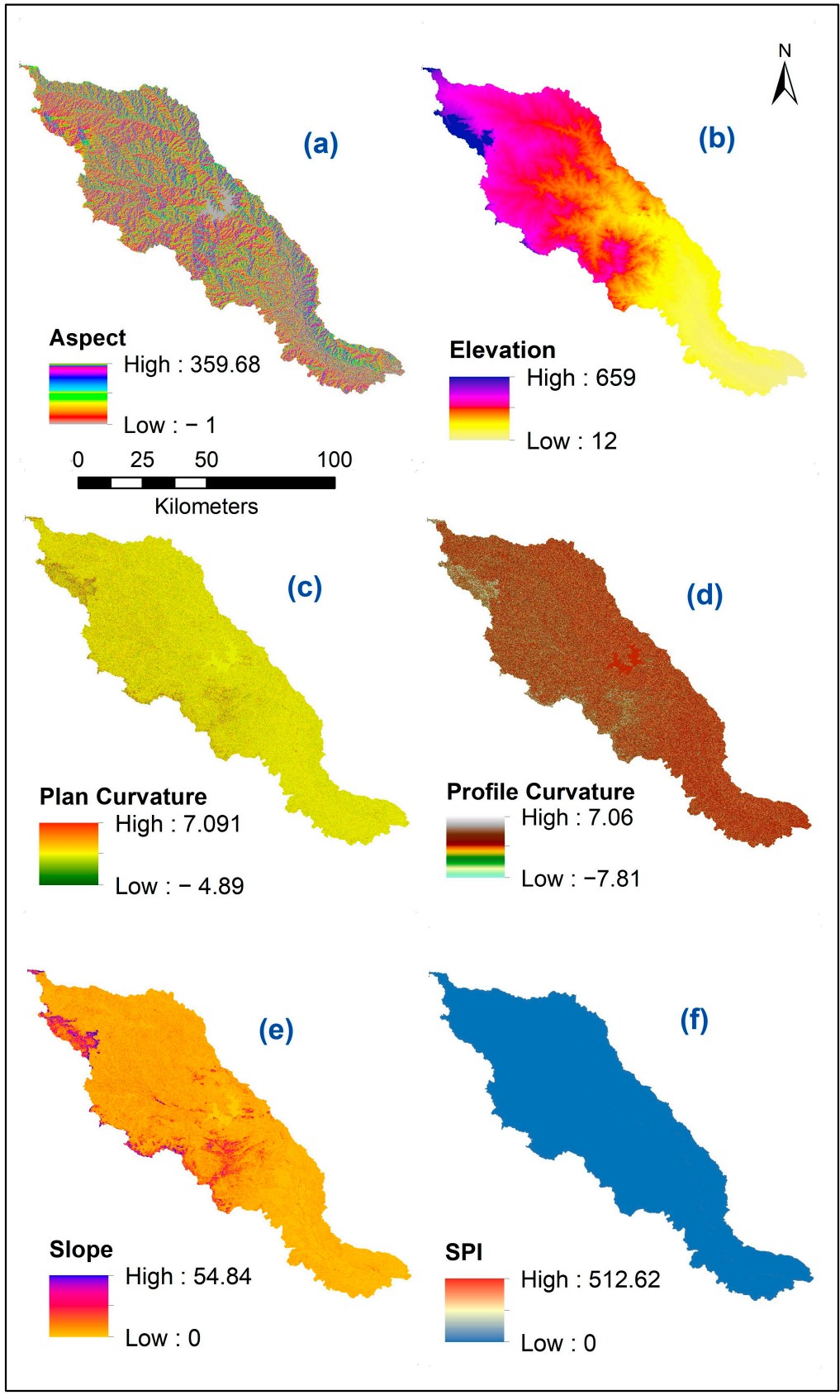

**Figure 3.** *Cont.*

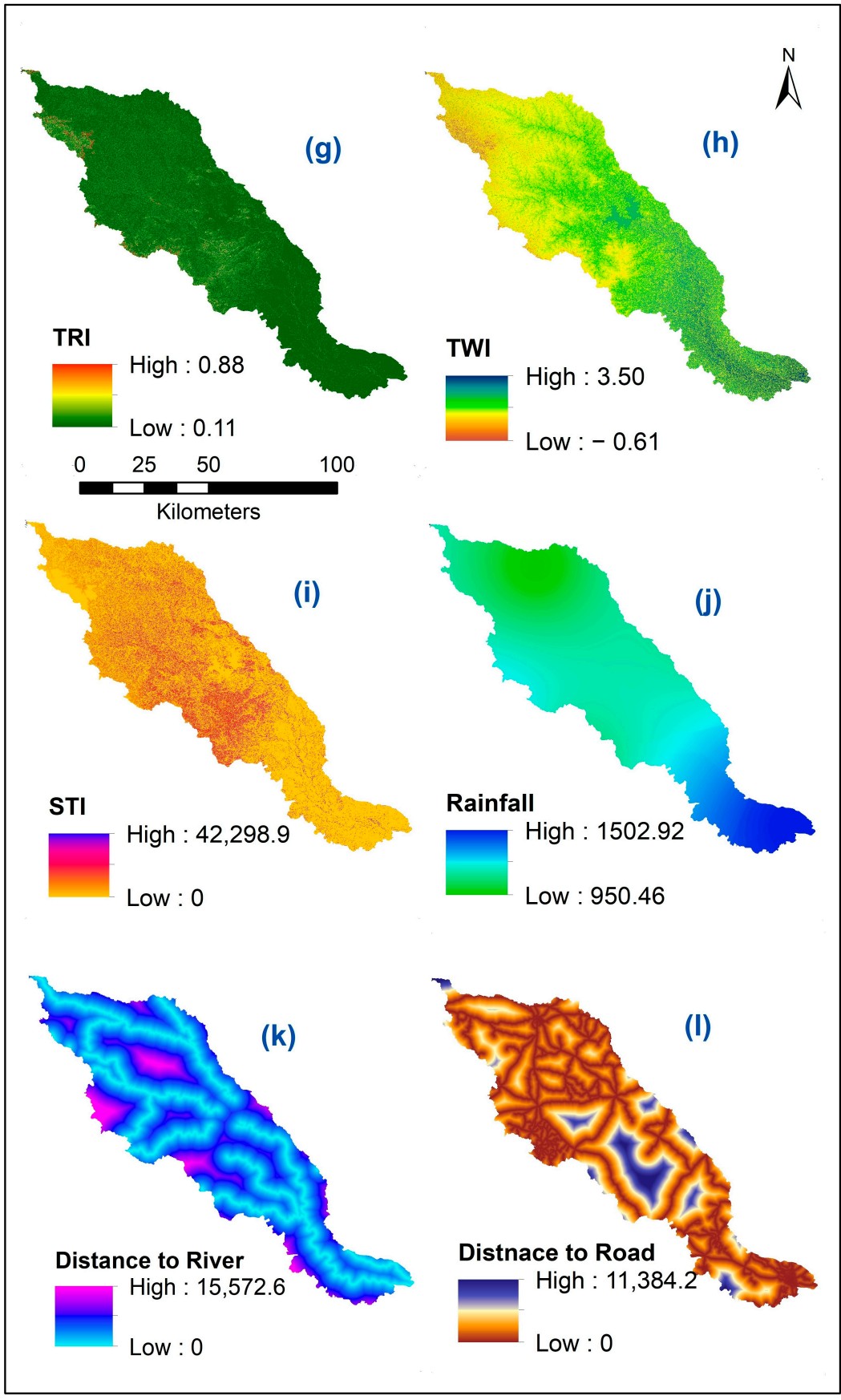

**Figure 3.** *Cont*.

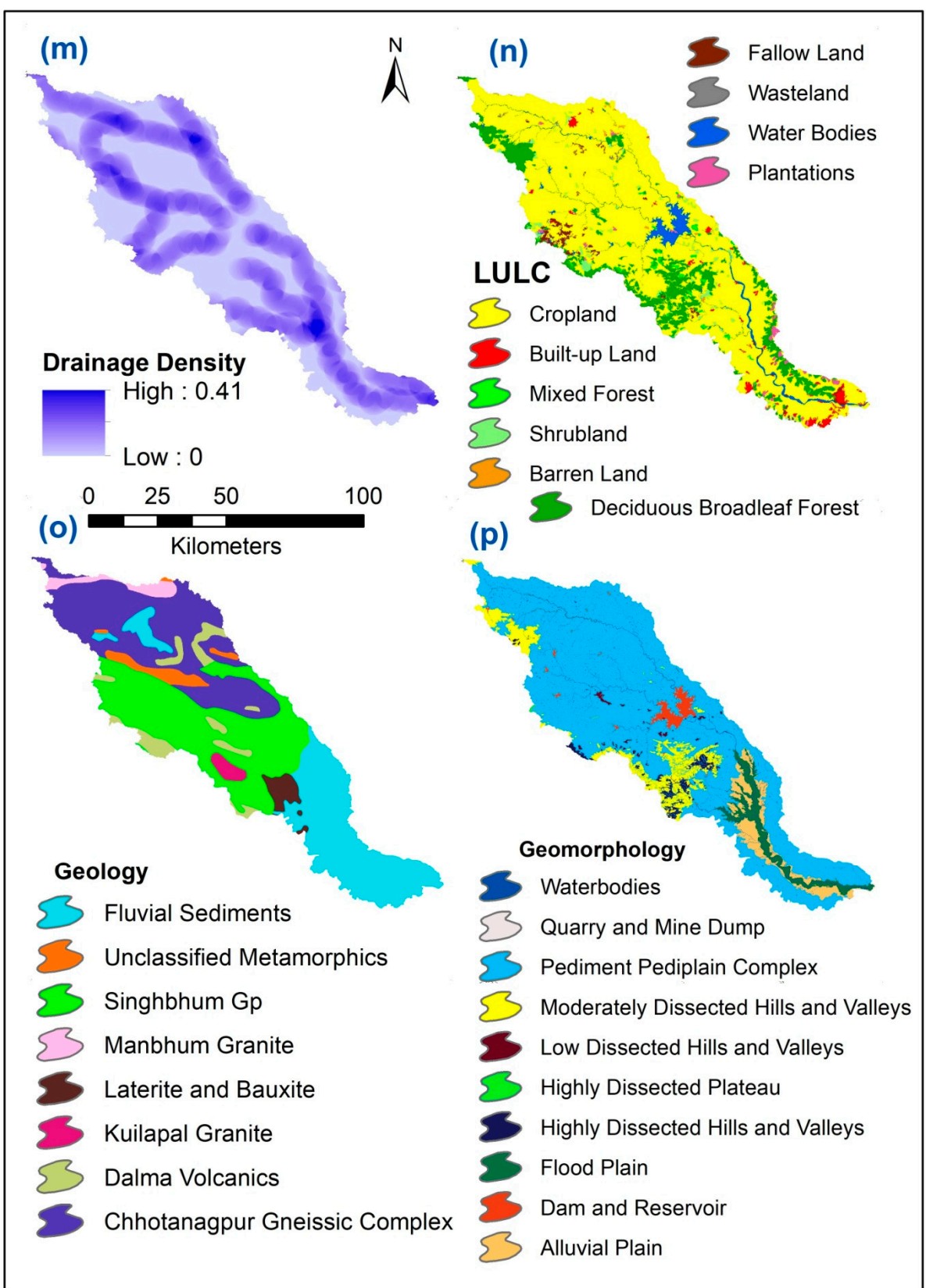

**Figure 3.** Flood causative factors: slope (**a**), elevation (**b**), plain curvature (**c**), profile curvature (**d**), slope (**e**), SPI (**f**), TRI (**g**), TWI (**h**), STI (**i**), rainfall (**j**), distance to river (**k**), distance to road (**l**), drainage density (**m**), LULC (**n**), geology (**o**) and geomorphology (**p**).

### 2.5.2. Random Forest (RF)

Breiman created the ensemble machine learning method random forest (RF), which predicts a model employing several classification or regression trees [43,44]. By using a "regression tree" the response variable was modeled in this study. It produces several suitable regression tree-based training models and after that, it categorizes the data in accordance with the voting results of numerous classifiers using sampling with replacement and diverse sets of the sample that were produced [45]. Once the prediction has the smallest node deviation achievable, the regression tree is trained to classify the observations based on the response variables [46]. A "nonlinear decision surface" was created by the "regression trees" rule, which is a collection of linear divisions of observed data. Because "regression trees" have a propensity in overfitting the training data, they function poorly when no data is supplied. This issue can be solved by a single regression tree trained with the help of the RF method. Randomly chosen records and predictor variables from the input data are used as input in the training section. Each set of "regression trees" and several sampling procedures result in the production of a collection of regression trees. The RF and generalization error are estimated using the following formulas [47]:

$$GE = P_{x,y}(mg(x,y) < 0) \tag{3}$$

$$mg(x,y) = av_k I(h_k(x) = y) - max_{j \neq y} av_k I(h_k(x) = j) \tag{4}$$

where "$x$ and $y$ indicates several factors to specify the likelihood over $x$ and $y$ space, $mg$ is to subsidiary task and $I(*)$ indicates the marker task".

### 2.5.3. Support Vector Machine (SVM)

Based on the "structural risk reduction" concept and "statistical learning theory" [48], a supervised ML algorithm called SVM was used. Through the creation of hyperplanes, the nonlinear properties of the world are reshaped into linearity, and SVM simplifies and processes it into classes [49]. The kernel function is the name for the mathematical procedure used to convert data. Using the training dataset, the original input is transformed by SVM into a high-dimensional feature space. The initial space between the points of two different classes is produced by hyper-plane separation and consists of $n$ coordinates ($x_i$ parameters in vector $x$). Marjanovi et al. (2011) [50] state that the classification "hyper-plane" is built by SVM in the center of the "maximum margin" of separation between the classes. The point will be classed as +1 if it is above the "hyper-plane", and $-1$ if it is not. The hyper-plane in an SVM model has been calculated using the following equations:

$$Min \sum_{i=1}^{n} \varphi_i - \frac{1}{2} \sum_{i=1}^{n} \sum_{j=1}^{n} \varphi_i \varphi_j y_i y_j (x_i, x_j) \tag{5}$$

Subject to:

$$Min \sum_{i=1}^{n} \varphi_i y_j = 0 \ and \ 0 \leq \alpha_i \leq D \tag{6}$$

where "$x = x_i$, $i = 1, 2, \ldots n$ stand for the key variables of vector, $y = y_i$, $j = 1, 2, \ldots n$ represent the output variables of vector and $\varphi_i$ is Lagrange multipliers".

The resulting task of SVM can be expressed as follows:

$$f(x) = sgn \left( \sum_{i=j}^{n} y_i \varphi_i K(x_i, x_j) + a \right) \tag{7}$$

where "a is the bias that indicates the hyperplane's linear distance from the origin, $K(x_i, x_j)$ is Polynomial (POL) and radial basis function (RBF) are examples of kernel functions, and they may be represented as follows":

$$K_{POL}(x_i, x_j) = ((x * y) + 1)^d \tag{8}$$

$$K_{RBF}(x_i, x_j) = e^{-y||x - x_i||^2} \tag{9}$$

### 2.5.4. Artificial Neural Network (ANN)

According to Luk, Ball, and Sharma (2001) [51], ANNs that mimic the human nervous system may create answers that are meaningful even when the input data is inaccurate or incomplete [52], it can also learn and generalize these data. A large variety of issues to be resolved [53], another acronym used is ANN. It has the capacity to reproduce a previously unrecognized association between a collection of "input elements (like rainfall) and output components (like runoff)" [54] or "groundwater level" [55]. To start computing the ANN, an array of numbers, xi, is initially introduced to the input layer of the processing nodes. Then, owing to "connection-specific weights", these signals may either be muted or amplified as they traverse links to each of the nodes in the subsequent layer. After being ignored by a "threshold function", the processing units subsequently convert the input signal into an output signal (Oj). The equation that follows is as follows:

$$f(x) = \frac{1}{1 + e^{-x}} \tag{10}$$

The derived result is computed as follows if the final results or output range from 0 to 1:

$$Qj = \frac{1}{1 + e^{-\sum x_1 w_1}} \tag{11}$$

"The output from the processing unit, f(x), ranges between 0 and 1".

### 2.5.5. Model Validation Techniques

Appropriate validation strategies are absolutely necessary for any scientific research. No research work can be applied in reality if the employed models have not been adequately validated [56,57]. Therefore, in our research, we have used the six most reliable statistical techniques namely "specificity, sensitivity, PPV, NPV, F score, andarea under the curve—receiver operating characteristics curve (AUC-ROC)"; in addition, one graphical technique, including Taylor diagram [58] (also used in this study), to accurately assess the model performance as well as predicted result. The validation approaches have been calculated using the four statistical metrics "TP, TN, FN, and FP". According to Pal et al., (2022) [59], the accuracy of the previously mentioned models is determined by these validation method values; the models performed better when the values were greater, and vice versa. The following equations were employed to perform all the validation measures:

$$AUC = \frac{(\sum TP + \sum TN)}{(P + N)} \tag{12}$$

$$Sensitivity = TP/(TP + FN) \tag{13}$$

$$Specificity = TN/(FP + TN) \tag{14}$$

$$PPV = \frac{TP}{FP + TP} \tag{15}$$

$$NPV = \frac{TN}{TN + FN} \tag{16}$$

$$Precession = \frac{TP}{TP + FP} \tag{17}$$

$$Recall = \frac{TP}{TP + FN} \tag{18}$$

$$F\ score = 2 * \frac{Precession * Recall}{Precession + Recall} \qquad (19)$$

## 3. Results

### 3.1. Multicollinearity Assessment

In our flood susceptibility assessment, the aforementioned MC analysis tools played a notable role in improving the modeling approach by eliminating the parameter bias and helping to select suitable flood conditioning factors. Therefore, sixteen variables are considered for flood susceptibility mapping with the help of MC assessment. In our present research, the VIF and TOL results show that there are no collinearity issues which range from 0.83–3.13 and 0.32–1.21, respectively (Table 1); so, the highest VIF value is 3.13 whereas the lowest TOL value is 0.32, which implies that all MC values of all 16 flood causative factors are within permissible limits.

**Table 1.** Multi-collinearity assessment of all considered parameters.

| Parameters | Multi-Collinearity | |
| --- | --- | --- |
| | TOL | VIF |
| Aspect | 0.89 | 1.12 |
| Elevation | 0.53 | 1.89 |
| Plan curvature | 1.21 | 0.83 |
| Profile curvature | 0.99 | 1.01 |
| Slope | 0.49 | 2.04 |
| SPI | 0.63 | 1.59 |
| TRI | 0.69 | 1.45 |
| TWI | 0.85 | 1.18 |
| STI | 0.77 | 1.30 |
| Rainfall | 0.78 | 1.28 |
| Distance to river | 0.52 | 1.92 |
| Distance to road | 0.87 | 1.15 |
| Drainage density | 0.32 | 3.13 |
| LULC | 0.67 | 1.49 |
| Geology | 0.71 | 1.41 |
| Geomorphology | 0.89 | 1.12 |

### 3.2. Flood Susceptibility Assessment

In our research work, RF, SVM, and ANN ML algorithms have been employed in delineating the FSZM with the help of all the adopted models, notably categorizing the entire Kangsabati river basin area into five distinctive zones including very low, low, moderate, high, and very high susceptibility areas for the well comprehended spatial distribution of flood-prone regions.

In the case of the SVM modeling approach the southern portion, the low-lying areas are significantly characterized by frequent flood occurrences (Figure 4a); therefore, the very high and high flood-prone regions that are found in the southern portion of this river basin contain 9.01% and 11.06%, respectively, of the total study area (Figure 5). Except for the southern portion, the remaining parts are specifically characterized by moderate (10.71%), low (22.01%), and very low (47.21%) regions.

In a very well-known ML algorithm, RF also shows remarkable results in flood susceptibility modeling (Figure 4b). In our study, the spatial distribution pattern is significant, which gives a somewhat similar result to the SVM modeling approach; whereas the very high (9.21%) and high (12.01%) flood zones are situated at the lower portion of the river basin that fall under the southern portion of considered regions (Figure 5). The northern and central regions are specifically associated with moderate (11.36%), low (24.17%), and very low (43.25%) flood susceptibility areas.

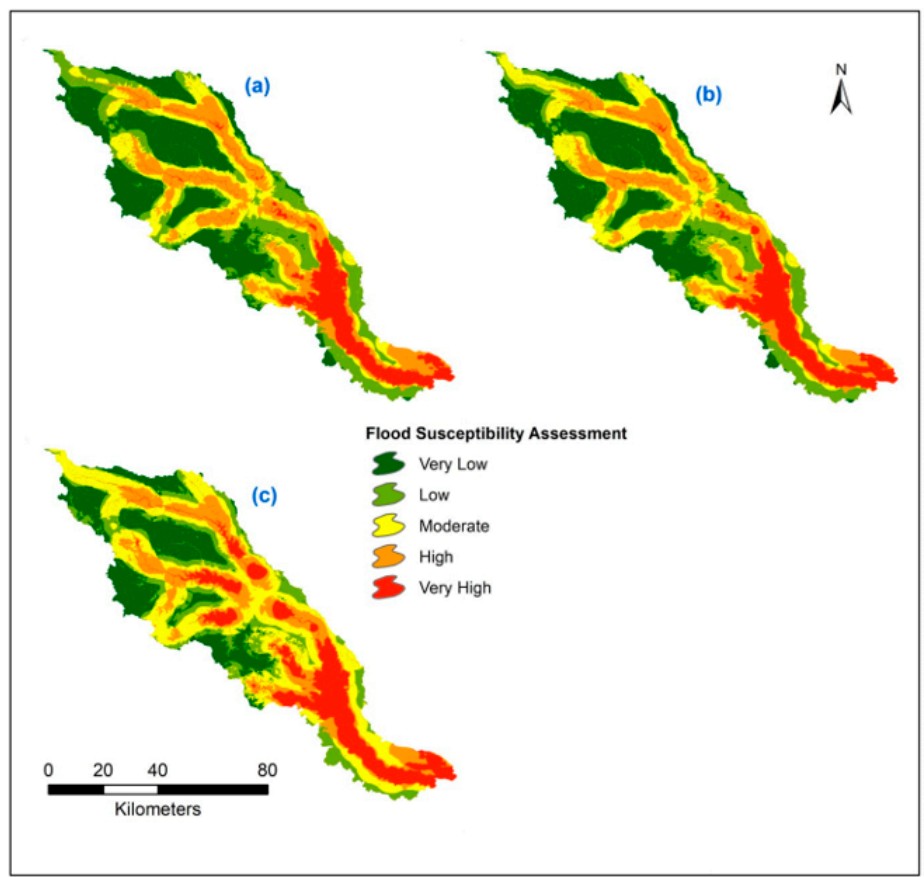

**Figure 4.** Flood susceptibility assessment using SVM (**a**), RF (**b**), and ANN (**c**) models.

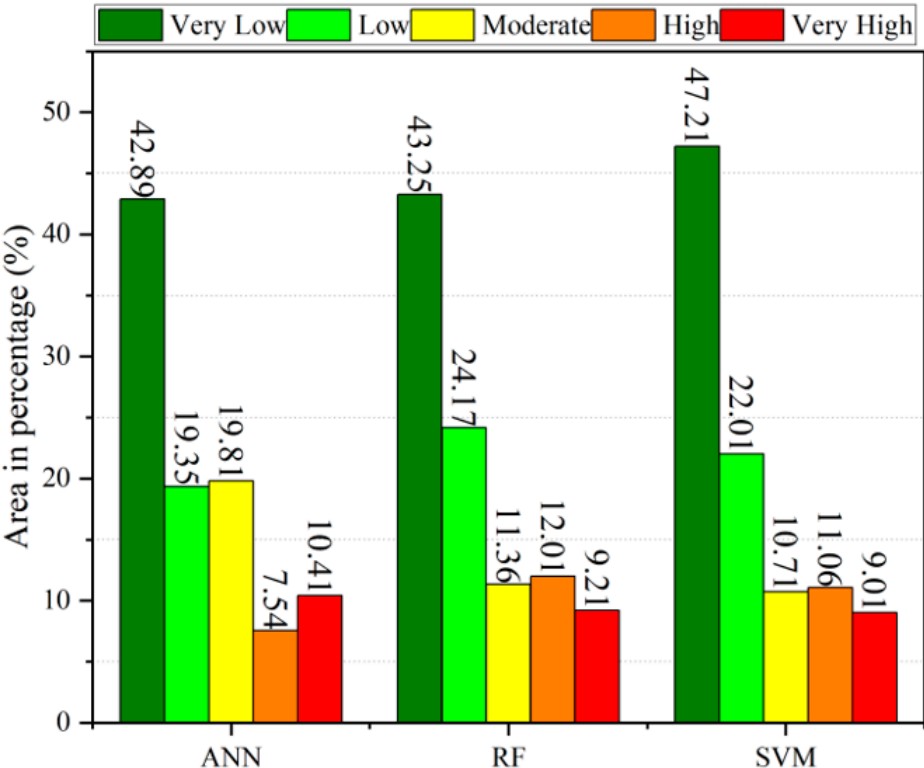

**Figure 5.** Areal coverage of flood susceptibility classes.

The FSZM is precisely developed by the ANN ML modeling approach, which gave noteworthy results (Figure 4c). The derived result shows that the south-central and southern part of the considered basin area notably fall under the very high to the high susceptible region, which covered 10.41% and 7.54%, respectively, of the entire region apart from those the northern and central 19.81%, 19.35% and 42.89% area come under the moderate, low, and very low flood susceptible zones, respectively (Figure 5). Therefore, the riparian residents who resided close to the river channel were severely affected by this flood.

### 3.3. Model Evaluation

Model evaluation is a very important part of all scientific studies. Therefore, the adopted models in our research work needed to be evaluated. In our current research work, the validation is much more important due to its numerous data sources and remotely sensed data, which may bring possible errors in research outcomes. Henceforth, Our models have been validated through the six most important and widely used statistical techniques, namely "specificity, sensitivity, PPV, NPV, F score, AUC-ROC", and one graphical technique, the Taylor diagram. All the validation techniques give noteworthy results that interpret that the model accuracy and precision level are significantly evaluated. In the present study, the result of AUC-ROC shown in Figure 6a (training) and Figure 6b (validating) shows that the ANN modeling approach (training—0.901; validating—0.891) gives a more accurate result followed by RF (training—0.880; validating—0.871) and SVM (training—0.835; validating—0.805) in both sections. Apart from this, all other adopted validation techniques also give similar results with AUC-ROC techniques, with ANN being the more prominent result compared with RF and SVM techniques. The validation measures of all adopted specificity ("training—0.86; validating—0.85"), sensitivity ("training—0.93; validating—0.92"), PPV ("training—0.85; validating—0.85"), NPV ("training—0.90; validating—0.91"), F score ("training—0.89; validating—0.88")techniques, and one graphical technique, the Taylor diagram (r = 0.88) (Figure 7), shows the more reliability of ANN model followed by RF and SVM (Table 2).

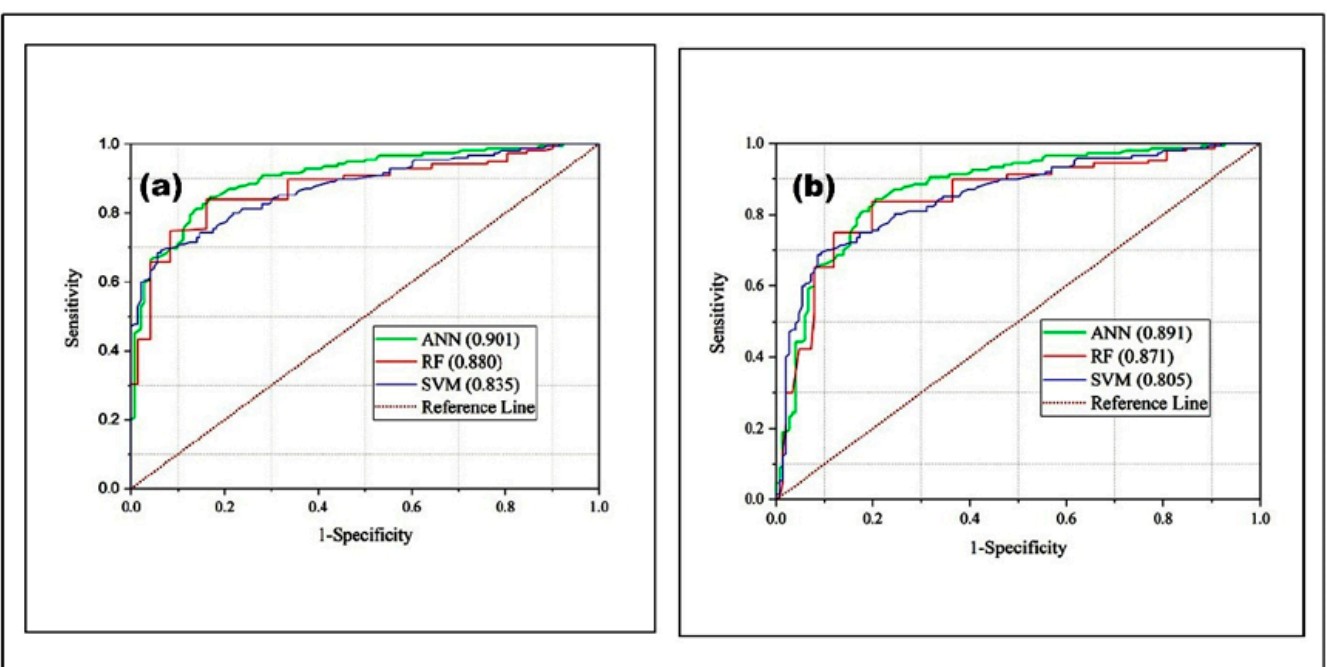

**Figure 6.** ROC curve using training (**a**) and validation datasets (**b**).

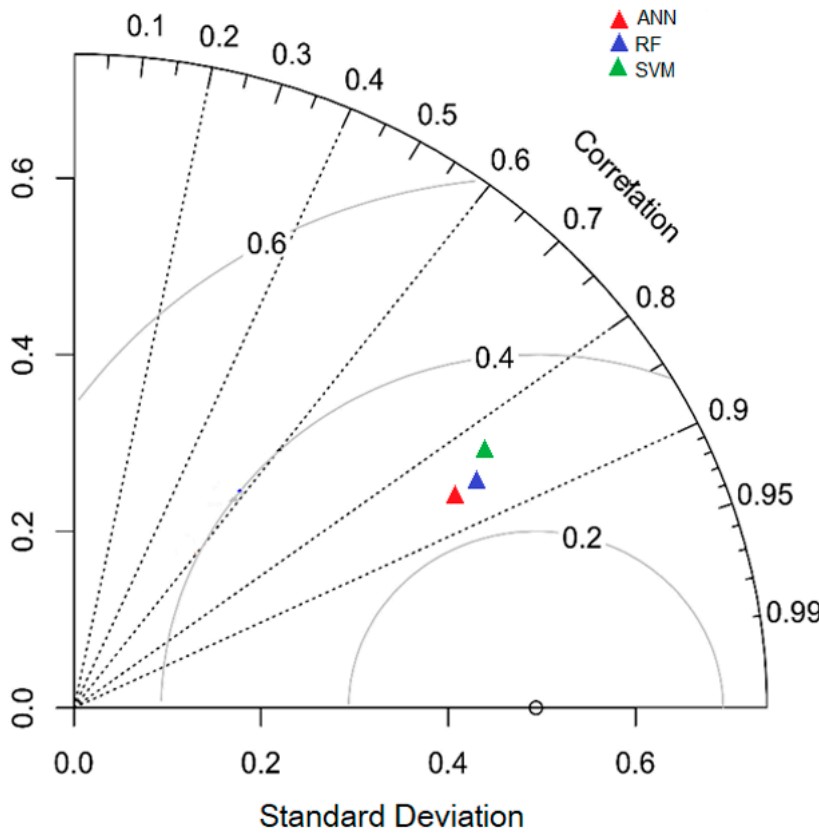

**Figure 7.** Degree of correspondence between modeled and observed outcomes of flood by the Taylor diagram.

**Table 2.** Validation of the models by using different validation measures.

| Models | Stage | Parameters | | | | | |
|---|---|---|---|---|---|---|---|
| | | Sensitivity | Specificity | PPV | NPV | F Score | AUC |
| ANN | Training | 0.93 | 0.86 | 0.85 | 0.90 | 0.89 | 0.901 |
| | Validation | 0.92 | 0.85 | 0.85 | 0.91 | 0.88 | 0.891 |
| RF | Training | 0.92 | 0.87 | 0.91 | 0.91 | 0.89 | 0.880 |
| | Validation | 0.91 | 0.82 | 0.93 | 0.92 | 0.86 | 0.871 |
| SVM | Training | 0.91 | 0.86 | 0.89 | 0.95 | 0.88 | 0.835 |
| | Validation | 0.89 | 0.83 | 0.86 | 0.97 | 0.86 | 0.805 |

## 4. Discussion

Flash floods are among the most severe and complicated natural disasters that may result in an immediate loss of property and human life due to their quick occurrence, fast water flow, and huge sediment transfer. However, full prevention of it is unfeasible. To protect lives and decrease the economic consequences of flood occurrences, which provide a variety of challenges for local government, flood prediction and mitigation strategies must be developed. A Flash Flood Guidance System with Global Coverage (FFGS), which has a hierarchical institutional framework, may provide alerts for South Asian nations including India, Nepal, Bhutan, Bangladesh, and Sri Lanka roughly six to twenty-four hours in advance. Since Bangladesh and India are now more densely populated than other mentioned countries, riverbank protection in those countries is much more deeply rooted at the state level, and powers are transferred to the grassroots level for implementation. However, since the subcontinent's legal and cultural foundations are still in place, execution and behavioral intention are still subject to some risk [7]. Due to the existence of various degrees of disaster-prone regions, India is one of the top ten most disaster-prone nations in

the world (Centre for Research on the Epidemiology of Disasters (CRED). Several areas have been susceptible to various natural dangers due to the geo-climatic conditions that exist in different portions of the nation. Other variables that also contribute to the rising catastrophe patterns in Indian states include global warming, faster population growth, rapid industrialization, urbanization, and illegal constructions [60]. One of the frequent occurrences of disasters in India is flooding. Various areas are very vulnerable to flood catastrophes. The severity of exposure to floods in Indian states has increased for a number of reasons, including high rain during monsoon seasons, a lack of river connection, rising urbanization, and unlawful development in urban areas with insufficient drainage and reservoir systems [61]. The incidence of flood catastrophe occurrences has an impact on both the socioeconomic well-being of the populace and the nation's economic progress [62]. The loss of life, damage to public and private property, and destruction of agricultural crops in several Indian states are the direct effects of flood catastrophes. Several efforts have been made by various researchers to develop an effective response plan; among them, FFZM is one of the key flood prevention measures that enable the quick identification of flood-susceptible areas as well as the adoption of appropriate and systemic procedures to reduce the consequences of flooding. Until now, various models and prediction techniques have been applied in delineating flood-prone areas, but the most suitable modeling approach for a particular place by considering all the geo-environmental factors with higher accuracy and reliability is crucial. Therefore, in our study, our foremost priority is to develop an accurate FSZM of the Kangsabati river basin with the help of the most prominent ANN, RF, and SVM modeling approaches. Henceforth, several studies [63–66] have been done in the RS-GIS environment, but in recent times ML algorithms with the help of artificial intelligence gain global acceptance in susceptibility modeling [67–71]. Based on the proper flood-impacting factors in a certain area, all of these strategies have produced the best results. Even though FSZM has made great advancements, more work must undoubtedly be done to improve flash flood susceptibility mapping's efficacy. Therefore, a thorough analysis of this difference is required to choose the right method for a certain study topic [72]. In flood-affected areas, zonation SVM is a significant tool in recent times. Wu et al., (2019) and Xiong et al., (2019) [73,74] have shown that SVM is the most reliable ML algorithm in flood-affected zonation mapping in China. A significant number of research works have been performed for flood-prone area mapping by using the RF modeling approach, acquiring very prominent results [75–77]. Apart from these models, ANN became the most used and reliable ML algorithm in prediction studies, not only in probable flood zonation but also in several hazard zonation mapping [78–82]. Falah et al., (2019) [83] and Shafizadeh-Moghadam et al., (2018) [84] have developed a modeling approach for Mashhad city and the Mazandaran province of Iran with the help of the ANN model, acquiring significant outcomes in flood hazard prediction.

According to Tehrany et al., (2015) [29], numerous geohydrological, morphological, and topographical factors contribute to floods. Although just a few elements have a major influence on the existence of flood events in a particular location, research on the flood-prone areas of Iran suggests that slope has a substantial effect [85] among different flood conditioning factors, whereas land use patterns are causing devastating floods in Vietnam [86]. In our present study, three models have shown that several flood causative factors, namely LULC, STI, distance to a river, and stream power index played crucial roles in the Kangsabati river basin. Besides flood-causing factors, model validation measures are also important, which can evaluate the adopted model. Likewise, in previous studies, these predictive measures demonstrate that ANN ML algorithms in our study also give more accurate results, followed by RF and SVM; the derived results are quite impressive and similar to previous research work. Several researchers such as Dahri et al., (2022) [87] and Samantaray et al., (2022) [88] show the wide acceptance of the ANN modeling approach for this higher predictive capability. Additionally, traditional statistical approaches need a lot of time, vast datasets, and additional input factors, making them inappropriate for areas with a lack of data, particularly emerging nations like India. Thus, the ANN model is

quite important for our study of FSZM in our study area due to the higher precision level compared with other adopted ML algorithms.

Firstly, the proposed method has some degree of uncertainty because there is not much knowledge available on flooding's hydrodynamics. Second, using the recommended method of applying the causative factors may be questionable due to the variable spatial resolution of the causal factors generated by digital elevation models (DEMs). Third, we limited our investigation to fluvial flash floods. Finally, based on expert evaluation and historical flood data, non-flood locations in the study basin were randomly selected, which might greatly distort the results. Therefore, to improve the quality of the input datasets, future studies should concentrate on the model that selects trustworthy non-flood locations.

The produced results and conclusions of the current study will assist administrators and researchers of flood hazards in assessing FSZM and in making decisions about how to manage and lessen the effects of flood events. The findings of our research also revealed the causes of flood events in the Kangsabati river basin. It suggests that when taking into account the geo-hydrological characteristics of a particular basin, in which hydrological data may be accessible, the suggested method may effectively estimate flood hazard zone in other basins. Additionally, it recommends that scholars use this aforementioned model to judge a region's sensitivity to flooding occurrences, and in the future, hydraulic models will be utilized to gauge the severity of the flooding in this area.

## 5. Conclusions

For mitigating the effects of floods, precise and trustworthy FSZM is important. Assessment of the risk of FSZM has been a hot topic both nationally and internationally in recent decades. Two significant problems that, in a regional context, contribute to an increase in flash floods are an anthropogenic invasion on river banks and climate change. In this study, we produced highly accurate flood-prone region predictions for a subtropical climate that lacked sufficient data. For this study, we employed the three most accepted distinctive ML algorithms worldwide, namely ANN, RF, and SVM in assessing the probable FSZ, with the help of sixteen available important flood causative factors; multi-collinearity tests by using two VIFs and tolerance techniques have a noteworthy role in flood condition factors selection. The developed predicted outcomes were also evaluated through seven validation measures: specificity, sensitivity, PPV, NPV, F-score Taylor diagram, and most importantly, AUC-ROC in both training and validation. The entire study shows that the central, south-central, and southern low-lying populated areas are significantly affected by frequent flood events, which implies about 10.41% and 7.54% of the entire river basin fall under very high and high FS, respectively. Therefore, among several factors, the increasing nature of LULC change and encroachment towards the river basin emerge as the most prominent factors for inundation in low-lying regions. All of the aforementioned validating techniques scientifically evaluated all the adopted predictive models and display that ANN is the most suitable model in this subtropical river basin. Henceforth, the derived map of the Kangsabati river basin can be used for better planning and taking proper preventive measures. In order to reduce flood impacts on human livelihoods and the local economy, hazard prevention and reduction measures are therefore required at the regional level. Therefore, it is important to educate the residents of flood-prone areas on the risk of flooding in a timely manner. Therefore, this FSZM is a crucial tool in reducing human and financial damages through the application of suitable management methods, which can aid the financial system in providing flood-affected areas with adequate compensation and enforcing the necessary regulations related to land use. The main task of future research is to conduct flood susceptibility research in various changing conditions of the environment. In this perspective, the uncertainty of the various climatic events should be incorporated to get more robust outcomes in changing climates.

**Author Contributions:** Conceptualization, S.C.P.; methodology, R.C. and P.R.; validation, R.C. and D.R.; data collocation, R.C.; writing—original draft preparation, P.R. and D.R.; writing—review and editing, R.C., P.R., D.R., A.S. and I.C.; visualization, R.C., P.R. and D.R.; supervision, S.C.P.; funding acquisition, S.C.P. All authors have read and agreed to the published version of the manuscript.

**Funding:** This research received no external funding.

**Institutional Review Board Statement:** Not applicable.

**Informed Consent Statement:** Not applicable.

**Data Availability Statement:** Not applicable.

**Conflicts of Interest:** The authors declare no conflict of interest.

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
