# Peer review of "Living with Floods Using State-of-the-Art and Geospatial Techniques: Flood Mitigation Alternatives, Management Measures, and Policy Recommendations"

_water, doi:10.3390/w15030558_

Round 1
Reviewer 1 Report
Reviewer’s Comments
Overall comment
This MS has the fair potential to publish in the Water after revision. The theme of the manuscript addressing Living with Floods Using State-of-the-Art and Geospatial Techniques: Flood Mitigation Alternatives, Management Measures, and Policy Recommendations The authors have made a good effort to study Flood Mitigation Alternatives, Management Measures, and Policy Recommendations. However, there is always room for improvement to raise the caliber of research, so nothing is ever final. The methodology section is not addressed properly. There is no review of the relevant literature (Literature Review) to highlight what approaches have already been employed in the study area. The authors should try to include this section.
I would like to thank the authors of the article for its fluent text. Throughout the article, I noticed some points that, if corrected, could enhance the article
Other comments
Revise your abstract.
* What are the key results?
* What are the practical implications of your research (how can the results be utilized by e.g., readers, community, or companies)?
Make a clear research motivation and why you think it is necessary to carry on this work. The research gaps need to be clearly articulated.
I assume that minor changes in the introduction section is necessary, but one of the important tasks after publishing a study is to increase its chance to be seen by the most possible number of researchers. So, the more you cite similar publication, the more the chance that the search engine in the publisher website propose your paper to the researcher. Besides of that, it will also complete your introduction section. As another advantage, it rises new ideas to the researchers by combining various methods or resolving drawback of one seen paper by reading the similar one or extending the methodology to a fully automatic one.
I Recommend some articles for your guideline and built your introduction section
Sohail, M.T., Hussan, A., Ehsan, M. et al. Groundwater budgeting of Nari and Gaj formations and groundwater mapping of Karachi, Pakistan. Appl Water Sci 12, 267 (2022). https://doi.org/10.1007/s13201-022-01795-0
Ran, J. & Nedovic-Budic, Z., 2016. Integrating spatial planning and flood risk management: A new conceptual framework for the spatially integrated policy infrastructure. Computers, Environment and Urban Systems, 57, 68-79 Available from: https://www.sciencedirect.com/science/article/pii/S0198971516300084.
Manzoor, Z., Ehsan, M., Khan, M.B., Manzoor, A., Akhter, M.M., Sohail, M.T., Hussain, A., Shafi, A., Abu-Alam, T. & Abioui, M., 2022. Floods and flood management and its socio-economic impact on Pakistan: A review of the empirical literature. Frontiers in Environmental Science, 10, 1-14 Available from: https://www.frontiersin.org/articles/10.3389/fenvs.2022.1021862.
1. Materials and methods
This section needs improvement, it is suggested to add some latest refences that support your methodology.
Figure 2. Methodology flowchart of this study.
On what basis you develop this figure, please explain it in proper way. It is requested to properly elaborate its methodology.
2.3. Flood Inventory Mapping
Why you used only three models (SVM, ANN, and RF)? Is there any specific reason of it?
2.5.1. Multicollinearity assessment
Please provide the reference of equation 1 and 2.
Table 1. Multi-collinearity assessment of all considered parameters.
On what’s basis you assign these values
What you conclude from this table
Discussions
What are your suggestions to improve the disaster management and quick response systems in south Asian countries, especially India?
It is suggested to write some discussion to improve Flood Warning System in India?.
Authors should make a comparative analysis of regional countries and present an effective solution that is vital to cater to such situations, (You can compare the policies of Bangladesh and Pakistan with India and identify the loopholes)
Author Response
Manuscript Title: Living with Floods Using State-of-the-Art and Geospatial Techniques: Flood Mitigation Alternatives, Management Measures, and Policy Recommendations
Manuscript ID: water-2161332
Reviewer’s Comments
Overall comment
This MS has the fair potential to publish in the Water after revision. The theme of the manuscript addressing Living with Floods Using State-of-the-Art and Geospatial Techniques: Flood Mitigation Alternatives, Management Measures, and Policy Recommendations The authors have made a good effort to study Flood Mitigation Alternatives, Management Measures, and Policy Recommendations. However, there is always room for improvement to raise the caliber of research, so nothing is ever final. The methodology section is not addressed properly. There is no review of the relevant literature (Literature Review) to highlight what approaches have already been employed in the study area. The authors should try to include this section.
I would like to thank the authors of the article for its fluent text. Throughout the article, I noticed some points that, if corrected, could enhance the article
Other comments
Revise your abstract.
* What are the key results?
* What are the practical implications of your research (how can the results be utilized by e.g., readers, community, or companies)?
Response
Thank you very much for your valuable observation, the Abstract section of the manuscript has been modified according to your suggestion.
Make a clear research motivation and why you think it is necessary to carry on this work. The research gaps need to be clearly articulated.
Response
Thank you very much for your valuable observation, the research gap of this work has been incorporated according to your suggestion.
I assume that minor changes in the introduction section is necessary, but one of the important tasks after publishing a study is to increase its chance to be seen by the most possible number of researchers. So, the more you cite similar publication, the more the chance that the search engine in the publisher website propose your paper to the researcher. Besides of that, it will also complete your introduction section. As another advantage, it rises new ideas to the researchers by combining various methods or resolving drawback of one seen paper by reading the similar one or extending the methodology to a fully automatic one.
I Recommend some articles for your guideline and built your introduction section
Sohail, M.T., Hussan, A., Ehsan, M. et al. Groundwater budgeting of Nari and Gaj formations and groundwater mapping of Karachi, Pakistan. Appl Water Sci 12, 267 (2022). https://doi.org/10.1007/s13201-022-01795-0
Ran, J. & Nedovic-Budic, Z., 2016. Integrating spatial planning and flood risk management: A new conceptual framework for the spatially integrated policy infrastructure. Computers, Environment and Urban Systems, 57, 68-79 Available from: https://www.sciencedirect.com/science/article/pii/S0198971516300084.
Manzoor, Z., Ehsan, M., Khan, M.B., Manzoor, A., Akhter, M.M., Sohail, M.T., Hussain, A., Shafi, A., Abu-Alam, T. & Abioui, M., 2022. Floods and flood management and its socio-economic impact on Pakistan: A review of the empirical literature. Frontiers in Environmental Science, 10, 1-14 Available from: https://www.frontiersin.org/articles/10.3389/fenvs.2022.1021862.
Response
Thank you very much for your valuable observation, we modified the Introduction section of the manuscript with considering your suggestion. In this perspective the similar articles have been cited. Apart from this the mentioned articles have been cited in proper part of the Introduction section with considering your suggestion.
- Materials and methods
This section needs improvement, it is suggested to add some latest refences that support your methodology.
Response
Thank you very much for your valuable observation, we modified the Materials and methods section considering your suggestion. In the same portion some latest references have also been incorporated.
Figure 2. Methodology flowchart of this study.
On what basis you develop this figure, please explain it in proper way. It is requested to properly elaborate its methodology.
Response
Thank you very much for your valuable observation, the basis for the development of the mentioned figures has been incorporated in the mentioned section of the manuscript. Apart from this the methodology of this work has been elaborated according to your suggestion.
2.3. Flood Inventory Mapping
Why you used only three models (SVM, ANN, and RF)? Is there any specific reason of it?
Response
Thank you very much for your valuable observation, the reason for the selection of the mentioned models has been incorporated in the manuscript.
2.5.1. Multicollinearity assessment
Please provide the reference of equation 1 and 2.
Response
Thank you very much for your valuable observation, the proper references of the mentioned equations has been incorporated.
Table 1. Multi-collinearity assessment of all considered parameters.
On what’s basis you assign these values
What you conclude from this table
Response
The VIF and TOL are two widely used statistics to test for multi-collinearity. Multi-collinearity occurs when values of specific predictors are spatially correlated with the values of another predictor. If VIF for a specific predictor is higher than 10 and TOL is lower than 0.1, then that variable should be removed from the modelling process due to high collinearity.
Discussions
What are your suggestions to improve the disaster management and quick response systems in south Asian countries, especially India?
It is suggested to write some discussion to improve Flood Warning System in India?.
Authors should make a comparative analysis of regional countries and present an effective solution that is vital to cater to such situations, (You can compare the policies of Bangladesh and Pakistan with India and identify the loopholes)
Response
Thank you very much for your suggestion; we have modified the entire Discussion section including all the suggestion from your end.

Reviewer 2 Report
After reviewing the current article, I found it interesting and easy to follow. It gives useful insights into flood susceptibility mapping, comparing three ML algorithms and validating the results with ground truth data. Also, the authors discuss the results and highlight their possible use in policymaking for flood mitigation. It is a quality paper. In the following bullets, I present some suggestions/revisions:
In the introduction it would be interesting to add some comments about flood aspects regarding the recent literature as follows.
· Line 43. The latest report of the IPCC also confirm the increase flood risk under future climate conditions (https://www.ipcc.ch/report/sixth-assessment-report-working-group-ii/)
· Line 50. Therefore the identification of flood exposure of critical/social infrastructure is necessary for rational flood risk management and prone areas prioritization. The ever-growing availability of EO data and well-establish use of GIS make feasible nowadays the assessment of flood exposure of infrastructures at national scale. To that end, some recent approaches were implemented lastly (https://doi.org/10.3390/hydrology9080145, https://doi.org/10.1016/j.ijdrr.2019.101240).
· Line 61. What about hydrological models uncertainties? Mention articles that try to evaluate models performance (https://link.springer.com/chapter/10.1007/978-3-030-02197-9_4, https://doi.org/10.1016/j.jhydrol.2021.126888)
· Line 72. Which was the research gap drive you to deal with that topic and which was your novel points in comparison other familiar approaches?
· Line 273. Please add a more comprehensive title for the figure 7. Taylor diagram about what?
· Line 275. Can you provide a ma representing the spatial errors of model performance?
· Provide some future research directions arise from this study.
Author Response
Manuscript Title: Living with Floods Using State-of-the-Art and Geospatial Techniques: Flood Mitigation Alternatives, Management Measures, and Policy Recommendations
Manuscript ID: water-2161332
After reviewing the current article, I found it interesting and easy to follow. It gives useful insights into flood susceptibility mapping, comparing three ML algorithms and validating the results with ground truth data. Also, the authors discuss the results and highlight their possible use in policymaking for flood mitigation. It is a quality paper. In the following bullets, I present some suggestions/revisions:
In the introduction it would be interesting to add some comments about flood aspects regarding the recent literature as follows.
- Line 43. The latest report of the IPCC also confirm the increase flood risk under future climate conditions (https://www.ipcc.ch/report/sixth-assessment-report-working-group-ii/)
- Line 50. Therefore the identification of flood exposure of critical/social infrastructure is necessary for rational flood risk management and prone areas prioritization. The ever-growing availability of EO data and well-establish use of GIS make feasible nowadays the assessment of flood exposure of infrastructures at national scale. To that end, some recent approaches were implemented lastly (https://doi.org/10.3390/hydrology9080145, https://doi.org/10.1016/j.ijdrr.2019.101240).
- Line 61. What about hydrological models uncertainties? Mention articles that try to evaluate models performance (https://link.springer.com/chapter/10.1007/978-3-030-02197-9_4, https://doi.org/10.1016/j.jhydrol.2021.126888)
Response
Thank you very much for your valuable observation, the overall introduction section has been modified with considering the recent literatures including the mentioned literatures form your end.
- Line 72. Which was the research gap drive you to deal with that topic and which was your novel points in comparison other familiar approaches?
Response
Thank you very much for your valuable observation, the research gap of this work has been incorporated according to your suggestion.
- Line 273. Please add a more comprehensive title for the figure 7. Taylor diagram about what?
Response
Thank you very much for your suggestion; the more comprehensive title of the mentioned Figure has been incorporated according to your suggestion.
- Line 275. Can you provide a ma representing the spatial errors of model performance?
Response
Thank you very much for your suggestion; the spatial errors of model performance have been incorporated according to the suggestion.
- Provide some future research directions arise from this study.
Response
Thank you very much for your suggestion; the future direction of this research has been incorporated in the mentioned section of the manuscript.

Round 2
Reviewer 1 Report
The authors significantly improve the paper in revision. I agree with revesion